# Beyond Grids: Learning Graph Representations for Visual Recognition

**Yin Li** *
Department of Biostatistics & Medical Informatics
Department of Computer Sciences
University of Wisconsin–Madison
`yin.li@wisc.edu`

**Abhinav Gupta**
The Robotics Institute
School of Computer Science
Carnegie Mellon University
`abhinavg@cs.cmu.edu`

## Abstract

We propose learning graph representations from 2D feature maps for visual recognition. Our method draws inspiration from region based recognition, and learns to transform a 2D image into a graph structure. The vertices of the graph define clusters of pixels ("regions"), and the edges measure the similarity between these clusters in a feature space. Our method further learns to propagate information across all vertices on the graph, and is able to project the learned graph representation back into 2D grids. Our graph representation facilitates reasoning beyond regular grids and can capture long range dependencies among regions. We demonstrate that our model can be trained from end-to-end, and is easily integrated into existing networks. Finally, we evaluate our method on three challenging recognition tasks: semantic segmentation, object detection and object instance segmentation. For all tasks, our method outperforms state-of-the-art methods.

## 1  Introduction

Deep convolutional networks have been tremendously successful for visual recognition [1]. These deep models stack many local operations of convolution and pooling. The assumption is that this stacking will not only provide a strong model for local patterns, but also create a large receptive field to capture long range dependencies, e.g., contextual relations between an object and other elements of the scene. However, this approach for modeling context is highly inefficient. A recent study [2] showed that even after hundreds of convolutions, the effective receptive field of a network's units is severely limited. Such a model may fail to incorporate global context beyond local regions.

Instead of the "deep stacking", one appealing idea is using image regions for context reasoning and visual recognition [3, 4, 5, 6, 7, 8, 9]. This paradigm builds on the theory of perceptual organization, and starts by grouping pixels into a small set of coherent regions. Recognition and context modeling are often postulated as an inference problem on a graph structure [8, 10]–with regions as vertices and the similarity between regions as edges. This graph thus encodes dependencies between regions. These dependencies are of much longer range than those are captured by local convolutions.

Inspired by region based recognition, we propose a novel approach for capturing long range dependencies using deep networks. Our key idea is to move beyond regular grids, and learn a graph representation for a 2D input image or feature map. This graph has its vertices defining clusters of pixels ("regions"), and its edges measuring the similarity between these clusters in a feature space. Our method further learns to propagate messages across *all* vertices on this graph, making it possible to share global information in a single operation. Finally, our method is able to project the learned graph representation back into 2D grids, and thus is fully compatible with existing networks.

Specifically, our method consists of *Graph Projection*, *Graph Convolution* and *Graph Re-projection*. Graph projection turns a 2D feature map into a graph, where pixels with similar features are assigned to the same vertex. It further encodes features for each vertex and computes an adjacency matrix for each sample. Graph convolution makes use of convolutions on a graph structure [11], and updates vertex features based on the adjacency matrix. Finally, graph re-projection interpolates the vertex features into a 2D feature map, by reverting the pixel-to-vertex assignment from the projection step.

We evaluate our method on several challenging visual recognition tasks, including semantic segmentation, object detection and object instance segmentation. Our method consistently improves state-of-the-art methods. For semantic segmentation, our method improves a baseline fully convolutional network [12] by ∼7%. And our results slightly outperform the state-of-the-art context modeling approaches [13, 14]. For object detection and instance segmentation, our method improves the strong baseline of Mask RCNN [15] by ∼1%. Note that a 1% improvement is significant on COCO (even doubling the number of layers provides 1-2% improvement). More importantly, we believe that our method offers a new perspective in designing deep models for visual recognition.

## 2 Related Work

Major progress has been made for visual recognition with the development of deep models. Deep networks have been widely used for image classification [1], semantic segmentation [12], object detection [16] and instance segmentation [15]. However, even after hundreds of convolution operations, these network may fail to capture long range context in the input image [2, 13].

Several recent works have thus developed deep architectures for modeling visual context. For example, dilated convolutions are attached to deep networks to increase the size of their receptive fields [17]. A global context vector, pooled from all spatial positions, can be concatenated to local features [18, 13]. These new features thus encode both global context and local appearance. Moreover, local features across different scales can be fused to encode global context [19]. However, all previous methods still reside in a regular 2D feature map with the exception of [20]. The non-local operation in [20] constructed a densely connected graph with pairwise edges between all pixels. Therefore, their method is computational heavy for high resolution feature maps, and is less desirable for tasks like semantic segmentation. Our methods differs from these approaches by moving beyond regular grids and learning an efficient graph representation with a small number of vertices.

Our method is inspired by region based recognition. This idea can date back to Gestalt school of visual perception. In this setting, recognition is posed as labeling image regions. Examples include segmentation [21], object recognition [3], object detection [22, 23] and scene geometry estimation [24]. Several works addressed context reasoning among regions. Context can be encoded via a decomposition of regions [6], or via features from neighborhood regions [7]. Our graph representation resembles the key idea of a region graph in [10, 9, 8], where vertices are regions and edges encode relationships between regions. While previous approaches did not consider deep models, our model embeds a region graph in a deep network. More recently, region based recognition has been revisited in deep learning [25, 26, 27]. Nonetheless, these methods considered grouping as a pre-processing step, and did not learn a graph representation as our method. In contrast, our method provides a novel deep model for learning graph representations of 2D visual data

Furthermore, our method is related to learning deep models on graph structure [28, 11, 29]. Specifically, graph convolutional networks [11] are used to propagate information on our learned graph. However, our method focuses on learning graph representations rather than developing message passing methods on the graph. Finally, our graph projection step draws inspirations from nonlinear feature encoding methods, such as VLAD and Fisher Vectors [30, 31]. These methods have been visited in the context of deep models [32, 33, 14]. However, previous methods focused on global encoding of local features, and did not consider the case of a graph representation.

## 3 Approach

In this section, we present our method on learning graph representations for visual recognition. We start with an overview of our key ideas, followed by a detailed derivation of the proposed graph convolutional unit. Finally, we discuss the learning of our method and present approaches for incorporating our model into existing networks for recognition tasks.

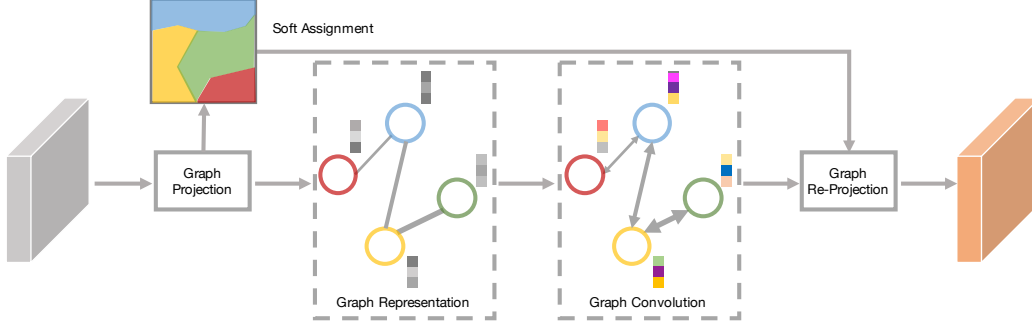

Figure 1: Overview of our approach. Our Graph Convolutional Unit (GCU) projects a 2D feature map into a sample-dependent graph structure by assigning pixels to the vertices of the graph. GCU then passes information along the edges of graph and update the vertex features. Finally, these new vertex features are projected back into 2D grids based on the pixel-to-vertex assignment. GCU learns to reason beyond regular grids, captures long range dependencies across the 2D plane, and can be easily integrated into existing networks for recognition tasks.

## 3.1 Overview

For simplicity, we consider an input 2D feature map $X$ of size $H \times W$ from a single sample. Our method can easily extend to batch size $\geq 1$ or 3D feature maps (e.g., videos). $X$ are the intermediate responses of a deep convolutional network. $x_{ij} \in R^d$ thus indexes the $d$ dimensional feature at pixel $(i, j)$. Our proposed Graph Convolutional Units (GCU) consists of three operations.

- **Graph Projection** $G_{proj}$. $G_{proj}$ projects $X$ into a graph $\mathcal{G} = (\mathcal{V}, \mathcal{E})$ with its vertices as $\mathcal{V}$ and edges as $\mathcal{E}$. Specifically, $G_{proj}$ assigns pixels with similar features to the same vertex. This assignment is soft and likely groups pixels into coherent regions. Pixel features are further aggregated within each vertex, and form the vertex features $Z \in R^{d \times |\mathcal{V}|}$ for graph $\mathcal{G}$. Based on $Z$, we measure the distance between vertices, and compute the adjacency matrix. Moreover, we store the pixel-to-vertex assignments and will use them to re-project the graph back to 2D grids.

- **Graph Convolution** $G_{conv}$. $G_{conv}$ performs convolutions on the graph $\mathcal{G}$ by propagating features $Z$ along the edges of the graph. $G_{conv}$ makes use of graph convolutions as [11] and can stack multiple convolutions with nonlinear activation functions. When $G$ is densely connected, $G_{conv}$ has a receptive field of *all vertices* on the graph, and thus is able to capture the global context of the input. $G_{conv}$ outputs the transformed vertex features $\tilde{Z} \in R^{|\mathcal{V}| \times \tilde{d}}$.

- **Graph Reprojection** $G_{reproj}$. $G_{reproj}$ maps the new features $\tilde{Z}$ back into the 2D grid of size $(H \times W)$. This is done by "inverting" the assignments from the projection step. The output $\tilde{X}$ of $G_{reproj}$ will be a 2D feature map with dimension $\tilde{d}$ at each position $(i, j)$. Thus, $\tilde{X}$ is compatible with a regular convolutional neural network.

Figure 1 presents an overview of our method. In a nutshell, our GCU can be expressed as

$$\tilde{X} = \text{GCU}(x) = G_{reproj}(G_{conv}(G_{proj}(X))). \tag{1}$$

It is more intuitive to consider our method in terms of pixels and regions. In GCU, "pixels" are assigned to vertices based on their feature vectors. Thus, each vertex defines a cluster of pixels, i.e., a "region" in the image. Each region will re-compute its feature by pooling over all its pixels. And the similarity between regions is estimated based on the pooled region features, and thus define the structure of a region graph. Inference can then be performed on the graph by passing messages between regions and along the edges that connect them. This inference will update the feature for each region and can connect regions that are far away in the 2D space. The updated region features can then be projected back to pixels by linearly interpolation between regions.

## 3.2 Graph Convolutional Unit

We now describe the details of our graph projection, convolution and reprojection operations.

**Graph Projection** $G_{proj}$ first assigns feature vectors $X$ to a set of vertices, parameterzied by $W \in R^{d \times |\mathcal{V}|}$ and $\Sigma \in R^{d \times |\mathcal{V}|}$, with the number of vertices $|\mathcal{V}|$ pre-specified. Each column $w_k \in R^d$ of $W$ specifies an anchor point for the vertex $k$. Specifically, we compute a soft-assignment $q_{ij}^k$ of a feature vector $x_{ij}$ to $w_k$ by

$$q_{ij}^k = \frac{\exp\left(-\|(x_{ij} - w_k)/\sigma_k\|_2^2/2\right)}{\sum_k \exp\left(-\|(x_{ij} - w_k)/\sigma_k\|_2^2/2\right)}, \tag{2}$$

where $\sigma_k$ is the column vector of $\Sigma$ and $/$ is the element-wise division. We constrain the range of each element in $\sigma_k$ to $(0, 1)$ by defining $\sigma_k$ as the output of a sigmoid function. Eq 2 computes the weighted Euclidean distance between all $x_{ij}$ and $w_k$, and creates a soft-assignment using softmax function. We denote $\mathcal{Q} \in R^{HW \times |\mathcal{V}|}$ as the soft assignment matrix from pixel to vertices, with each row vector $q_{ij}$ such that $\sum_k q_{ij}^k = 1$.

Moreover, we encode features $z_k$ for each vertex $k$ by

$$z_k = \frac{z_k'}{\|z_k'\|_2}, \quad z_k' = \frac{1}{\sum_{ij} q_{ij}^k} \sum_{ij} q_{ij}^k (x_{ij} - w_k)/\sigma_k. \tag{3}$$

Each $z_k'$ is a weighted average of the residuals between feature vectors $x_{ij}$ to the vertex parameter $w_k$. $z_k'$ is further L2 normalized to get the feature vector $z_k$ for vertex $k$. $z_k$ thus forms the $k$th columns of the feature matrix $Z \in R^{d \times |\mathcal{V}|}$. We further compute the graph adjacency matrix as $\mathcal{A} = Z^T Z$. With normalized $z_k$, $\mathcal{A}_{k,k'}$ in the adjacency matrix is the pairwise cosine similarity between the feature vectors $z_k$ and $z_{k'}$. Note that removing the coefficients $1/\sum_{ij} q_{ij}^k$ does not impact the normalized feature $z_k$, yet will change the way that the gradients are computed.

Eq 3 is inspired by nonlinear feature encoding methods, such as VLAD or Fisher Vectors [30, 34, 31]. This connection is more obvious if we consider $w_k$ as the cluster center and $\sigma_k$ as its variance (assuming a diagonal covariance matrix). In this case, the L2 normalization is exactly the intra-normalization in [34]. We note that our encoding is different from VLAD or Fisher Vectors as we do not concatenate $z_k$ as a global representation of $X$. Instead, we derive graph structure from $Z$ and keep individual $z_k$ as vertex features.

Eq 3 can be viewed as multiple parallel yet competing affine transforms, followed by weighted average pooling. And thus each $z_k$ provides a different snapshot of the input $X$. Moreover, if $w_k$ and $\sigma_k$ are computed as per batch mean and variance for cluster $k$, each affine transform becomes batch normalization [35]. This link between fisher vector and batch normalization is discussed in [36].

To summarize, the outputs of our graph projection operation are (1) the adjacency matrix $\mathcal{A}$, (2) the vertex features $Z$ and (3) the pixel-to-vertex assignment matrix $\mathcal{Q}$. Moreover, our graph projection operation introduce $2|\mathcal{V}|d$ new parameters. With a small number of vertices (e.g., 32) and a moderate feature dimension (e.g., 1024), the number of added parameters is small in comparison to those in the rest of a deep network. Moreover, every step in this project operation is fully differentiable. Thus, chain rule can be used for the derivatives of the input $x$ and the parameters ($W$ and $\Sigma$). In practice, we reply on automatic differentiation for back propagation.

**Graph Convolution** We make use of graph convolution $G_{conv}$ from [11] to further propagate information on the graph. Specifically, for a single graph convolution with its parameter $W_g \in R^{d \times \tilde{d}}$, the operation is defined as

$$\tilde{Z} = f(\mathcal{A} Z^T W_g), \tag{4}$$

where $f$ can be any nonlinear activation functions. We use the Batch Normalization [35] with Rectified Linear Units for our models. For all our experiments, we use a single graph convolution yet stacking multiple graph convolutions is a trivial extension. While each graph convolution has parameters of size ($d \times \tilde{d}$), it remains highly efficient with a small number of vertices. Note that our adjacency matrix is computed per sample, and thus our graph representation is sample-dependent and will get updated during training. This is different from the settings in [28, 11, 29], where a sample-independent graph is pre-computed and remains unchanged during training.

**Graph Reprojection** Our graph reprojection operation $G_{reproj}$ takes the inputs of transformed vertex features $\tilde{Z}$ and the assignment matrix $\mathcal{Q}$, and produces 2D feature map $\tilde{X}$. Ideally, we have to invert the assignment matrix $\mathcal{Q}$, which is unfortunately unfeasible. Instead, we compute pixel features

of $\tilde{X}$ using a re-weighting of the vertex features $\tilde{Z}$, given by $\tilde{X} = \mathcal{Q}\tilde{Z}^T$. $G_{reproj}$ thus linearly interpolates 2D pixel features based on their region assignments and does not have any parameters. Note that even if two pixels are assigned to the same vertex, they will have different features after reprojection. Thus, GCU is likely to preserve the spatial details of the signal. Finally, these projection results can be integrated into existing networks.

### 3.3 Learning Graph Representations

Our model is fully differentiable and can be trained from end-to-end. However, we find that learning GCUs faces an optimization challenge. Let us consider a corner case where the model assigns most of the input pixel features $x_{ij}$ to a single vertex $k$. In this setting, the GCU will degenerate to a linear function $W_g(x_{ij} - w_k)/\sigma_k$. As this rare case seems to be unlikely, we find that the model can be trapped to several modes. For example, the model will always assign the whole image with a single vertex, but uses different vertices for different images. To address this issue, we propose two strategies to regularize the learning of GCU.

**Initialization by Clustering** We initialize the $W$ and $\Sigma$ in graph projection operations by clustering the input feature maps. Specifically, we use K-Means clustering to get the centers for each column $w_k$ of $W$. We also estimate the variance along each dimension and setup the column vectors $\sigma_k$ of $\Sigma$. Note that our model always start with a pre-trained network. Thus, K-Means will produce semantically meaningful clusters. And this initialization does not require labeled data. Once initialized, we use gradient descent to update $W$ and $\Sigma$, and avoid tracking batch statistics as [35]. We found that this initialization is helpful for stable training and gives slightly better results than random initialization.

**Regularization by Diversity** We find it beneficial to directly regularize the assignment between pixels to vertices. Specifically, we propose to add a graph diversity loss function that matches the distribution of the assignments $p(\mathcal{Q}) = \sum_{ij} q_{ij}^k \in R^{|\mathcal{V}|}$ to a prior $p$. This is given by

$$\mathcal{L}_{div} = KL(p(\mathcal{Q})||p) \tag{5}$$

where $KL$ is the Kullback–Leibler divergence between $p(\mathcal{Q})$ and $p$. This regularization term is highly flexible as it allows us to inject any prior distributions. We assume that $p$ follows a uniform distribution. This prior enforces that each vertex is used with equal frequency, and thus prevents the learning of "empty" vertices. We find that a small coefficient (0.05) of this loss term is sufficient.

### 3.4 Graph Blocks

Our GCU can be easily wrapped into a graph block and incorporated into an existing network. Specifically, we define a graph block as

$$\hat{X} = X \oplus \text{GCU}_{k_1}(X) \oplus ... \oplus \text{GCU}_{k_n}(X), \tag{6}$$

where $\text{GCU}_k$ denote a GCU with $k$ vertices and $\oplus$ can be either a residual connection or a concatenation operation. A residual connection allows us to insert GCU into a network without changing its behavior (by using a zero initialization of the batch normalization after $G_{conv}$). Concatenating features supplements the original map $X$ with global context captured at different granularity. We explore both architectures in our experiments. We use the residual connection with a single GCU for object detection and instance segmentation, and concatenate features from multiple GCUs for semantic segmentation. Details of these architectures are shown in Fig 2.

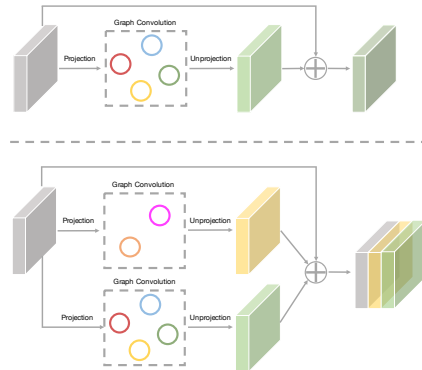

Figure 2: Architectures of two different graph blocks. Top: Single GCU with a residual connection can be incorporated an existing network; Bottom: Concatenation of multiple parallel GCUs introduces a new context model.

**Computational Complexity** We summarize the computational complexity of our graph block. More importantly, we compare the complexity to 2D convolutions and non-local networks [20].

- Complexity: For a feature map of size $H \times W$ with dimension $d$, our graph projection has a complexity of $O(HWd|\mathcal{V}|)$, where $|\mathcal{V}|$ is the number of vertices on the graph. The graph convolution is $O(|\mathcal{V}|d^2 + |\mathcal{V}|^2 d)$ and the reprojection takes $O(HWd|\mathcal{V}|)$ if we keep the output feature dimension the same as the inputs.

- Comparison to convolutions. The graph projection, convolution and reprojection operations have roughly the same complexity as 1x1 convolutions with output dimension $|\mathcal{V}|$ (assuming $HW \geq d$ for high resolution feature maps). Thus, the complexity of a single GCU is approximately equivalent to the stacking of three 1x1 convolutions.

- Comparison to non-local networks. For the same setting, the non-local operation [20] has a complexity of $O(H^2W^2d)$. With a high resolution feature map (large $HW$) and a small $|\mathcal{V}|$, this is almost quadratic to our GCU.

# 4 Experiments

We present our experiments and discuss the results in this section. We test our method on three important recognition tasks: (1) semantic segmentation, (2) object detection and instance segmentation. Our experiments are thus organized into two parts.

We also explored different ways of incorporating our method in the experiments. For semantic segmentation, we concatenate multiple GCU outputs (Fig 2 Bottom). In this case, our method has to be accomplished with extra convolutions for recognition, and thus can be considered an novel context model. For object detection and instance segmentation, we incorporate GCUs with residual connections (Fig 2 Top) into the Mask RCNN framework [15]. Here, our method does not change the original networks and thus serves as a new plugin unit.

## 4.1 Semantic Segmentation

We now present our results on semantic segmentation. We introduce the benchmark and implementation details, and present an ablation study of the GCU. More importantly, we compare our model to a set of baselines and discuss the results.

**Dataset and Benchmark** We use ADE20K dataset [37] for semantic segmentation. ADE20K contains 22K densely labeled images. The benchmark includes 150 semantic categories with both stuff (i.e., wall, sky) and objects (i.e., car, person). The categories are fine-grained and their number of samples follows a long tailed distribution. Therefore, this dataset is very challenging. We follow the same evaluation protocol as [13] and train our method on the 20K training set. We report the pixel level accuracy and mean Intersection over Union (mIoU) on the 2K validation set.

**Implementation Details** Our base model attaches 4 GCUs to the last block of a backbone network and concatenates their outputs, followed by convolutions for pixel labeling. These GCUs have $(2, 4, 8, 32)$ vertices and output dimensions of $\tilde{d} = 256$. And these numbers are chosen to roughly match the number of parameters and operations as in [13]. We use ResNet 50/101 [38] pre-trained on ImageNet [39] as our backbone network. Similar to [13], we add dilation to the last two residual blocks, thus the output is down-sampled by a factor of 8. We upsample the result to original resolution using bilinear interpolation. As in [13], we crop the image into a fixed size (505x505) with data augmentations (random flip, rotation, scale) and train for 120 epochs. We also add an auxiliary loss after the 4th residual block with a weight of $0.4$ as [13]. The network is trained using SGD with batch size 16 (across 4 GPUs), learning rate $0.01$ and momentum $0.9$. We also adapt the power decay for learning rate schedule [40], and enable synchronized batch normalization. For inference, we average network outputs from multiple scales.

**Ablation Study** We provide an ablation study of the GCU using the task of semantic segmentation. The results are reported on ADE20K and with pre-trained ResNet 50 as the backbone. First, we vary the number of GCUs. With dilated convolutions, the backbone itself has a mIoU of $35.6\%$. Adding a single GCU with 2 vertices to the backbone achieves $39.43\%$–a $\sim 4\%$ boost. Using two GCUs with $(2, 4)$ vertices reaches $40.92\%$. And our base model (4 GCUs with $(2, 4, 8, 32)$ vertices) has $42.60\%$. Alternatively, if we increase the number of vertices in the last GCU of our base model (from 32 to 64). The mIoU score stays similar to the base model ($42.58\%$ vs. $42.60\%$). Adding more nodes increases

Figure 3: Visualization of segmentation results on ADE20K (with ResNet 50). Our method produces "smoother" maps–regions that are similar are likely to be labeled as the same category.

| Backbone | Method | PixAcc% | mIoU% |
|---|---|---|---|
| VGG16 [42] | FCN-8s [12] | 71.32 | 29.39 |
| | SegNet [41] | 71.00 | 21.64 |
| | DilatedNet [17] | 73.55 | 32.31 |
| | CascadeNet [37] | 74.52 | 34.90 |
| Res50 [38] | Dilated FCN | 76.51 | 35.60 |
| | PSPNet [13] | 80.76 | **42.78** |
| | EncNet [14] | 79.73 | 41.11 |
| | GCU (ours) | 79.51 | **42.60** |
| Res101 [38] | RefineNet [19] | - | 40.20 |
| | PSPNet [13] | 81.39 | 43.29 |
| | EncNet [14] | 81.69 | **44.65** |
| | GCU (ours) | 81.19 | **44.81** |

Table 1: Results of semantic segmentation on ADE20K. mIoU scores within $0.5\%$ of the best result are marked. With ResNet 50, our method improves Dilated FCN by $7\%$. With ResNet 101, our method outperforms PSPNet by $1.5\%$.

the run-time and memory cost, yet does not seem to improve the performance. Second, we evaluate our initialization and regularization schemes. Our base model (4 GCUs) without regularization and initialization has a mIoU of $41.34$. Our regularization improves the result by $0.39\%$ ($41.73\%$). Our initialization further adds another $0.87\%$ ($42.60\%$). Thus, both the diversity loss and the clustering help to improve the performance.

**Baselines** We further compare our method with a set of baselines. These baselines are organized as

- Dilated FCN: This is the backbone network of our method, where we added dilation to a ResNet. It is also a variant of DeepLab [40].

- Context Models: We include results from recent context models for deep networks. Specifically, we compare to state-of-the-art results from PSPNet [13], RefineNet [19] and EncNet [14]. These are close competitors of our method.

- Other Methods: We also report results from [12, 17, 41, 37, 19] for reference.

**Results and Discussions** Our main results are summarized in Table 1. Our method (GCU) improves the backbone Dilated FCN network by $7\%$ in mIoU. With ResNet 50, our result on mIoU is comparable to PSPNet. With ResNet 101, our method is $1.5\%$ better than PSPNet and $4.6\%$ higher than RefineNet in mIoU. We also notice that our pixel level accuracy is consistently lower than PSPNet by $0.2\text{-}1.2\%$. One possibility is that GCU will produce "diffused" pixel features. This is because the output features of GCU are linearly interpolated from region features, which are averaged across pixels. We visualize our results in Fig 3 and find that our method does tend to over-smooth the outputs (see the missing clock in the left column). A similar property is also observed in region based recognition [10]. Even a good grouping may decrease the performance if the recognition goes wrong. For example, in the middle column of Fig 3, our method mis-classified the "building" region as "house" and has a lower score than the baseline Dilated FCN. Nonetheless, our method is able to assign the same category to the pixels on the building surface, which are previously divided in pieces.

Thus far, we have described our method by taking the analogy of region based recognition. However, we must point out that our method is trained without supervision of regions. And there is no guarantee that it will learn a valid representation of regions or region graphs. To further diagnose our method, we create visualizations of the assignment matrix in GCU (see Fig 4). It is interesting to see that our method does learn to identify some meaningful components of the scene. For example, with 2 vertices, the network seems to build up the concept of foreground vs. background. With 4 vertices, there seems to be a weak correlation between the assignments and the spatial layout (e.g., pink for ground, and yellow for vertical surfaces). As the number of vertices grows, the assignment begins to over-segment the image, creating superpixel-like regions.

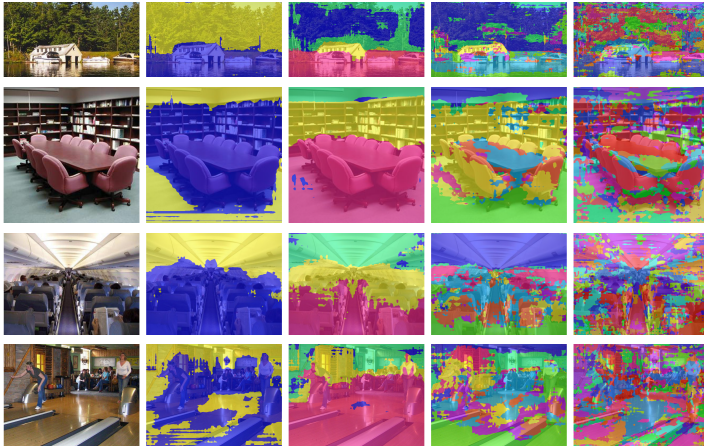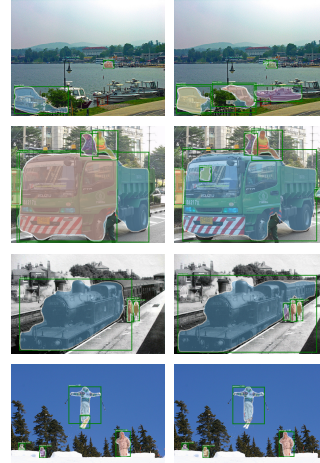

Figure 4: Visualization of the assignment matrix in GCU for semantic segmentation (with ResNet 50). From left to right: input image, pixel-to-vertex assignments with 2, 4, 8 and 32 vertices. Pixels with the same color are assigned to the same vertex. Vertices are colored consistently across images.

Figure 5: Visualization of object instance segmentation results (with ResNet 50). Left: Mask RCNN; Right: Ours. Zoom in for details.

| Backbone | Method | $AP^{box}$ | $AP_{50}^{box}$ | $AP_{75}^{box}$ | $AP^{seg}$ | $AP_{50}^{seg}$ | $AP_{75}^{seg}$ |
|---|---|---|---|---|---|---|---|
| ResNet 50 [38] | Mask RCNN [15, 20] | 38.0 | 59.6 | 41.0 | 34.6 | 56.4 | 36.5 |
| | Mask RCNN + NL [20] | **39.0** | 61.1 | 41.9 | **35.5** | 58.0 | 37.4 |
| | Mask RCNN(Detectron) [15, 44] | 37.7 | 59.2 | 40.9 | 33.9 | 55.8 | 35.8 |
| | Mask RCNN(Detectron) + GCU | **38.7** | 60.5 | 41.7 | 34.7 | 57.2 | 36.5 |
| ResNet 101 [38] | Mask RCNN [15, 20] | 39.5 | 61.3 | 42.9 | 36.0 | 58.1 | 38.3 |
| | Mask RCNN + NL [20] | **40.8** | 63.1 | 44.5 | **37.1** | 59.9 | 39.2 |
| | Mask RCNN(Detectron) [15, 44] | 40.0 | 61.8 | 43.7 | 35.9 | 58.3 | 38.0 |
| | Mask RCNN(Detectron) + GCU | **41.1** | 63.2 | 44.9 | **36.9** | 59.8 | 39.0 |

Table 2: Results of object detection and instance segmentation on COCO dataset. We compare our *single-model* results to state-of-the-art methods on COCO `minival`. Scores of $AP^{box}$ and $AP^{seg}$ with in 0.5% of the best result are marked. Our method (GCU) improves the strong baseline of Mask RCNN by ~1% across different networks.

## 4.2 Object Detection and Instance Segmentation

We now present our benchmark and results on object detection and segmentation.

**Dataset and Benchmark** For both object detection and instance segmentation, we use COCO dataset from [43]. COCO is by far the most challenging dataset for object detection and instance segmentation. COCO includes more than 160K images, where object bounding boxes and masks are annotated. We report the standard COCO metrics including AP (averaged over IoU thresholds), and $AP_{50}$, $AP_{75}$ (AP at different IoU thresholds) for both boxes and masks. As in previous work [15, 16], we train using the union of 80k train images and a 35k subset of val images (trainval35k), and report results on the remaining 5k val images (minival).

**Implementation Details** For both tasks, we attach 4 GCUs to the last four residual blocks in ResNet 50/101 [38] with FPN [45]. These GCUs have 32 vertices and output dimensions $\tilde{d} = [256, 512, 1024, 2048]$ that matches the feature dimensions of the network. Our GCUs are added with residual connections and zero initialization after the last convolution of the residual block and before FPN layers. We train the model using SGD with a batch size of 8 across 4 GPUs. Following the training schedule (x1) in [44], we linearly scale the training iterations (180K) and initial learning rate (0.01) based on our batch size. The learning rate is decreased by 10 at 120/160K iterations. We also

freeze the batch normalization layers. Other implementation details for training and inference are kept the same as [15]. Note that only random flip is used for data augmentation during training. And our results are reported without test time augmentation (e.g., multi-scale or flip). They can be further incorporated for to improve performance. It is also possible to further boost the performance for the baseline and our method by training for longer (as the x2 scheme in [44]).

**Baselines** Our method is a plugin unit for Mask RCNN. Our baselines thus include

- Mask RCNN: This is the result reported in [20]. This version is slightly better than the original Mask RCNN [15] by replacing the stage-wise training with end-to-end training.
- Mask RCNN + NL: This is the result of adding non-local operations to the backbone network of Mask RCNN [20]. This operation is designed to capture long range dependencies.
- Detectron: This is the open source version of Mask RCNN [44] with end-to-end training and careful learning rate schedule. Our method builds on top of this implementation.

**Results and Discussions** Our results for both tasks are summarized in Table 2. Our method consistently improves the baseline Mask RCNN (Detectron) results by $\sim 1\%$ for both detection and segmentation, and for both ResNet 50 and 101. This trend of improvement is also observed by adding non-local networks. We have to emphasis that our baseline (Detectron) is a well-optimized version of Mask RCNN. Thus, our improvements are non-trivial. Moreover, we present visualizations of our results and compare them to the Mask RCNN (Detectron) in Fig 5. By modeling context using a graph representation, our method is able to find objects that are previously missing (the "boat" in first row), resolve ambiguity in region classification ("truck" vs "bus" in second row) and help to better estimate the spatial extent of objects (third row). One of the failure modes of our model is the missing detection of small, out-of-context objects, such as the "skis" in the sky (zoom in to see in last row). We hypothesis that this is again due to the "diffused" local features in GCU.

## 5   Conclusion

In this paper, we have presented a novel deep model for learning graph representations from 2D visual data. Our method transforms a 2D feature map into a graph structure, where the vertices define regions and edges capture the relationship between regions. Context modeling and recognition can be done using this graph structure. In this case, our method resembles the key idea behind region based recognition. Our model thus addresses pixel grouping, region representation, context modeling and recognition under the same framework. We have evaluated our method on several challenging visual recognition tasks. Our results outperformed state-of-the-art methods. Through careful analysis of these results, we demonstrated that our model is able to learn primitive grouping of scene components (such as foreground vs. background), and further leverage these components for recognition. Our method thus provides a revisit to the region based recognition in the era of deep learning. We hope our work will offer useful insights in re-thinking the design of visual representations in deep models.

**Acknowledgments**   This work was supported by ONR MURI N000141612007, Sloan Fellowship, Okawa Fellowship and ONR Young Investigator Award to AG. The authors thank Xiaolong Wang for many helpful discussions, and Jianping Shi for sharing implementation details of PSPNet.

## Footnotes

*The work was done when Y. Li was at CMU.

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
