[Reviews · NeurIPS 2018]

Reviewer 1



The paper proposes to learn graph representations from visual data via graph convolutional unit (GCU). It transforms a 2D feature maps extracted from a neural network to a sample-dependent graph, where pixels with similar features form a vertex and edges measure affinity of vertices in a feature space. Then graph convolutions are applied to pass information along the edges of the graph and update the vertex features. Finally, the updated vertex features are projected back to 2D grids based on the pixel-to-vertex assignment. GCU can be integrated into existing networks allowing end-to-end training and capturing long-range dependencies among regions (vertices). The approach is evaluated on ADE20k for semantic segmentation task building upon PSPNet with ResNet-50/100 backbone and on MS COCO for object detection and instance segmentation tasks using Mask RCNN architecture. The method consistently improves over considered baselines (PSPNet, Mask RCNN) across different tasks. Pros: - A novel model has been presented for learning graph representations from 2D data. - The proposed method seems to be effective and efficient, allows end-to-end learning and can be integrated into existing networks. Cons: - The number of vertices has to be pre-specified. For efficiency reasons the number of vertices is kept small (set up to max 32 in the experiments). This results in very coarse, overly smooth outputs and missed finer details/objects (lines 240, 294), and will not be suitable for the applications that require pixel-accurate predictions. - No ablation study is provided. The paper lacks analysis of contributions of different steps to the performance, e.g. how much does initialization by clustering and regularization described in sec. 3.3. affect the final results? Also many design choices are not justified/analyzed in the paper, e.g. number of GCUs and number of vertices. Why were different number of vertices chosen for different task? How does the number of GCUs / number of vertices influence the results? - The learnt groupings of pixels does not form meaningful semantic regions. Minor comments/typos: - Table 3,4 should be changed to Figure 3,4 - Line 293: mode -> modes Overall, the paper presents a novel approach for learning graph representations from 2D data, which seems to be effective and flexible. However, it lacks the analysis and ablation study of different components. This work does not provide any insights on the design choices and the reader would have troubles transferring the proposed method to other tasks or new architectures. In addition, the proposed approach has a shortcoming that the number of vertices has to be pre-specified which negatively influences the quality of output predictions.

Reviewer 2



The authors propose to learn graph representation from 2D feature maps by applying a sequence of operations, including projection, graph convolution, and reprojection. The proposed graph representation is able to encode longer range context information. The effectiveness of the proposed method has been shown on three challenging tasks: semantic segmentation, object detection and instance segmentation. Strengths: - A novel framework that captures long-range context information and can be trained end-to-end. - Proposed models show improvements on three challenging tasks: semantic segmentation, object detection, and instance segmentation. - Provided informative analysis about the learned graph representations for semantic segmentation. Weakness: - Some training details are missing. It may be better to provide more details in order for others to reproduce the results. In particular, a. K-Means is applied to initialize the centers. However, in the beginning of training stage, the feature maps are essentially random values. Or, is K-Means applied to a pretrained model? b. KL divergence between p(Q) and p is included in the loss. May be interesting to see the final learned p(Q) compared to p. c. It is not clear to the reviewer how one could backpropagte to x, w, and \sigma in Equation (2). Also, will the training be sensitive to \sigma? - Is it possible to visualize how the graph representations evolve during training? - What is the extra cost in terms of computation resources and speed when employing the GCU? - 4 GCUs with (2, 4, 8, 32) vertices are used for semantic segmentation. It may be informative to discuss the improvement of each added GCU (e.g., which one is more helpful for segmentation). - It may be also informative to provide the learned graph representations for object detection and instance segmentation. - The proposed model attains similar performance as non-local neural network. It may be informative to provide some more discussion about this (e.g., extra computational cost for each method.). - In line 121, what does \sigma_k = sigmoid(\sigma_k) mean? \sigma_k appears on both sides. In short, the reviewer thinks the proposed method is very interesting since it is able to capture the long-range context information for several computer vision tasks. It will be more informative if the authors could elaborate on the issues listed above.

Reviewer 3



# Summary The work tackles the important problem of graph based representations for CNNs in the context of computer vision problems. It is a direct application of a convolution technique presented before in combination with an approach to map image data to and and from graphs. The work presents strong results for semantic segmentation, object detection and object instance segmentation. # Paper Strengths - Relevant problem. - Fairly easy idea and combination of relevant work. - Improved prediction performance on several hard computer vision tasks. - Paper well written. # Paper Weaknesses - The presented node count for the graphs is quite low. How is performance affected if the count is increased? In the example of semantic segmentation: how does it affect the number of predicted classes? - Ablation study: how much of the learned pixel to node association is responsible for the performance boost. Previous work has also shown in the past that super-pixel based prediction is powerful and fast, I.e. with fixed associations. # Typos - Line 36: and computes *an* adjacency matrix - Line 255: there seems to be *a weak* correlation # Further Questions - Is there an advantage in speed in replacing some of the intermediate layers with this type of convolutional blocks? - Any ideas on how to derive the number of nodes for the graph? Any intuition on how this number regularises the predictor? - As far as I can tell the projection and re-projection is using activations from the previous layer both as feature (the where it will be mapped) and as data (the what will be mapped). Have you thought about deriving different features based on the activations; maybe also changing the dimension of the features through a non-linearity? Also concatenating hand-crafted features (or a learned derived value thereof), e.g., location, might lead to a stronger notion of "regions" as pointed out in the discussion about the result of semantic segmentation. - The paper opens that learning long-range dependencies is important for powerful predictors. In the example of semantic segmentation I can see that this is actually happening, e.g., in the visualisations in table 3; but I am not sure if it is fully required. Probably the truth lies somewhere in between and I miss a discussion about this. If no form of locality with respect to the 2d image space is encoded in the graph structure, I suspect that prediction suddenly depends on the image size.